# Inhibiting Fatty Acid Oxidation Suppresses Acquired Resistance to Standard Chemotherapy in Melanoma

**DOI:** 10.3390/ijms26209873

**Published:** 2025-10-10

**Authors:** Wonyoung Choi, Woojin Ham, Jeong Hwan Park, Sung Hoon Sim, Jung Won Chun, Mingyu Kang, Chaeyoung Kim, Woosol Hong, Eun-Byeol Koh, Joon Hee Kang, Sang Myung Woo, Soo-Youl Kim

**Affiliations:** 1Department of Cancer Biomedical Science, National Cancer Center Graduate School of Cancer Science and Policy, Goyang 10408, Republic of Korea; wonyoungchoi@ncc.re.kr (W.C.); deli4927@ncc.re.kr (J.W.C.); wnsl2820@gmail.com (J.H.K.); wsm@ncc.re.kr (S.M.W.); 2Division of Cancer Biology, Research Institute of National Cancer Center, Goyang 10408, Republic of Korea; 76681@ncc.re.kr (W.H.); 77423@ncc.re.kr (J.H.P.); kmg@ncc.re.kr (M.K.); 77060@ncc.re.kr (C.K.); 76824@ncc.re.kr (W.H.); 77512@ncc.re.kr (E.-B.K.); 3Therapeutic Resistance Research Branch, Research Institute and Hospital, National Cancer Center, Goyang 10408, Republic of Korea; 4New Cancer Cure-Bio Co., Goyang 10408, Republic of Korea; 5Division of Rare and Refractory Cancer, Research Institute, National Cancer Center, Goyang 10408, Republic of Korea; simsh@ncc.re.kr; 6Division of Clinical Research, Research Institute, National Cancer Center, Goyang 10408, Republic of Korea; 7Center for Liver and Pancreatobiliary Cancer, National Cancer Center, Goyang 10408, Republic of Korea

**Keywords:** melanoma, Dacarbazine, Dabrafenib, fatty acid oxidation, KN510713

## Abstract

Immunotherapy and RAF-targeted therapy have become standard treatments for melanoma, significantly improving outcomes compared to earlier therapies. When resistance to initial treatment develops, the older chemotherapy drug Dacarbazine is used. However, resistance to both therapies has emerged, promoting ongoing research to further enhance survival rates. Among various theories, autophagy is believed to play a critical role in acquired drug resistance, as increased autophagy has been observed in resistance to multiple anticancer agents. In this study, Dabrafenib was administered to melanoma cells with an RAF mutation, while Dacarbazine was given to cells with an Raf wild type. Both cell lines showed increased autophagy and FAO following treatment with the anticancer drugs. When FAO was blocked during drug treatment, melanoma cells became more susceptible to cell death. In xenograft models, B16F10 melanoma (Raf wild type) demonstrated regrowth due to acquired resistance after two weeks of Dacarbazine treatment. Conversely, a combination of Dacarbazine and the FAO inhibitors KN510 and KN713 (a combination of KN510 and KN713:KN510713) caused near-complete remission without regrowth. A375 melanoma (*BRAF^V600E^*) developed resistance after four weeks of Dabrafenib treatment, yet the combination of Dabrafenib and KN510713 resulted in near-complete remission with no signs of regrowth. Based on these findings, combining FAO inhibitors with first-line therapies may be a promising approach for managing melanoma, regardless of RAF mutation status.

## 1. Introduction

The systemic treatment landscape of malignant melanoma has undergone a profound transformation over the past decade. Clinical trials have demonstrated substantial improvements in overall survival with the use of anti–PD-1 therapies, either as monotherapy or in combination with anti–CTLA-4 agents [1,2]. In patients harboring *BRAF* mutations, dual inhibition of BRAF and MEK has also significantly prolonged survival [3]. With the advent of these novel therapeutic strategies, the median overall survival for patients with metastatic melanoma has increased from less than one year during the era of cytotoxic chemotherapy to more than three to five years in the current era of immune checkpoint inhibitors and targeted therapies [4,5]. Based on the 2018 WHO genomic classification criteria, melanoma is categorized into three groups determined by the intensity of chronic ultraviolet radiation exposure/cumulative solar damage [6]. However, to date, immunotherapy is used regardless of *BRAF* mutation status. For patients with *BRAF* mutations, combination therapy with BRAF and MEK inhibitors is employed [7]. Following the FDA approval of ipilimumab [an antibody targeting the cytotoxic T lymphocyte-associated protein 4 (CTLA4)] for treating metastatic melanoma in 2011 [8], immune checkpoint inhibitor therapy has emerged as a first-line immunotherapy for melanoma by targeting CTLA-4 and PD-1 on T cells or programmed cell death ligand 1 (PD-L1) on tumor cells to prevent anti-tumor responses [7,9]. BRAF inhibitors like Vemurafenib and Dabrafenib are used alone or in combination with MEK inhibitors to treat patients with *BRAF*-mutated metastatic melanoma [10,11]. Most patients exhibit a rapid response to these drugs, which are now a vital part of treatment; however, acquired resistance often develops [12]. On the other hand, Dacarbazine is the most effective cytotoxic agent for treating advanced metastatic melanoma. It has been the standard chemotherapy for this malignant tumor for over 30 years and is still used as a subsequent treatment option when first-line treatment fails [13,14]. Dacarbazine (5-(3,3-dimethyl-1-triazeno) imidazole-4-carboxamide; DTIC) causes DNA damage by methylating nucleic acids, leading to cell growth arrest and cell death [15]. Unfortunately, only 5–10% of patients with metastatic melanoma respond to Dacarbazine, and most relapse within a few weeks [16]. Therefore, drug resistance in melanoma develops quickly and strongly [17]. Furthermore, the responses and prognosis with immune checkpoint inhibitors tend to be less favorable in Asian patients, underscoring an even greater unmet clinical need [18]. To enhance the effectiveness of anticancer therapy for melanoma, it is essential to develop strategies that overcome drug resistance.

Autophagy has already been suggested as a factor that contributes to drug resistance in different types of cancer [19]. In the early stages of melanoma development, autophagy was often seen as a process that suppresses tumor growth [19]. However, during cancer progression, including melanoma, autophagy has been linked to tumor-promoting functions [19,20]. Autophagy significantly increased in a xenograft model with aggressive melanoma compared to indolent melanoma [21]. Therefore, inhibiting autophagy combined with temozolomide treatment led to a significant increase in cell death in aggressive melanoma spheroid culture [21]. Furthermore, high expression of the autophagy marker LC3-II was associated with tumor malignancy and poorer chemotherapy outcomes [22]. It is often observed that melanoma with *BRAF* mutations develops resistance to BRAF inhibitor treatments. BRAF inhibitor-resistant tumors exhibited increased levels of autophagy compared to controls [23]. Patients with higher levels of chemotherapy-induced autophagy showed significantly lower response rates to chemotherapy and had a shorter survival time [23]. Therefore, combining autophagy inhibition with BRAF inhibition enhanced the anticancer effect on melanoma xenografts using BRAF inhibitor-resistant melanoma [23]. Results from a recently published phase 1/2 multicenter clinical trial showed that the combination therapy of Dabrafenib, trametinib, and the autophagy inhibitor hydroxychloroquine was safe and effective in patients with *BRAF*-mutated melanoma [24,25]. Dacarbazine also triggers autophagy, which might play a role in chemotherapy resistance [26]. This suggests that blocking autophagy could enhance the efficacy of Dacarbazine.

In this study, we examined common metabolic targets that could reduce resistance to standard treatments, such as Dabrafenib and Dacarbazine, in melanoma. We observed that autophagy and fatty acid oxidation (FAO) increase when melanoma is treated with chemotherapeutic agents, such as Dabrafenib or Dacarbazine. We investigated whether acquired drug resistance in melanoma is closely linked to the rise in FAO through autophagy. Therefore, we also assessed whether inhibiting FAO suppresses the development of acquired drug resistance in melanoma xenograft models.

## 2. Results

### 2.1. FAO Was Activated in Melanoma and Was Enhanced by Chemotherapy Treatment

ACAA1 (acetyl-CoA acyltransferase1) expression was confirmed through immunohistochemical staining of tissue microarrays (TMAs) containing normal skin samples (*n* = 30) and melanoma tissues from patients (*n* = 163) (Appendix A). ACAA1 has been reported to have higher expression in pancreatic cancer than in normal tissue [27]. Representative images are shown; scale bars are black (200 µm) and blue (50 µm). H-scores calculated with Inform software (version 2.6.0) indicated median ACAA1 scores of 45.3 in normal skin and 166.1 in melanoma tissue (Figure 1A). Analysis of the TCGA dataset also showed that eight FAO-related genes were listed. We confirmed that both *BRAF* mutant and wild-type melanomas had significantly higher levels compared to normal tissue (Appendix A). To assess whether FAO activity increases in melanoma, BSA-conjugated palmitic acid (C16:0) was applied to B16F10 (Braf wild type), A375 (*BRAF^V600E^*), UACC-257 (*BRAF^V600E^*), and SK-MEL-5 cells (*BRAF^V600E^*) for 3 h, followed by OCR analysis (Figure 1B–E). XF Seahorse analysis revealed approximately 18–35% higher ATP production compared to controls, along with a significant increase in basal respiration (Figure 1B–E and Appendix A). To validate this in multiple cancer types, experiments were conducted using PDAC, NSCLC, liver cancer, GBM, and breast cancer cells, demonstrating a notable increase in ATP output. No change was observed in the normal pancreatic cell line HPNE (Appendix A). We examined whether FAO increased over time at doses of anticancer drugs that allowed melanoma cell survival rather than death. First, we determined the concentration that enabled cell survival using the SRB assay (Appendix A). Treatments included Dacarbazine (200 µM) on wild-type melanoma B16F10 cells and Dabrafenib (50 nM) on *BRAF^V600E^* mutant melanoma A375 cells. After 0–72 h of treatment, FAO-related protein levels of carnitine-acylcarnitine carrier (CAC) and ACAA1 increased in a time-dependent manner—for example, CAC increased 5.8-fold in B16F10 and 11.2-fold in A375 after 72 h (Figure 1F). Recently, we have reported that CAC expression is higher in pancreatic cancer than in normal tissue [28]. Additionally, the expression of OxPhos Complexes I–V was significantly higher, and FAO protein levels also increased following Dabrafenib treatment in other melanoma cell lines, UACC-257 and SK-MEL-5 (Appendix A). After Dacarbazine and Dabrafenib treatment, ATP levels increased by 33% and 45%, respectively, at 72 h compared to baseline (Figure 1G). These findings suggest that melanoma cells depend on FAO as a primary energy source to survive under conditions of Dacarbazine or Dabrafenib treatment.

### 2.2. FAO Inhibition Reduced the Reduction of Energy Cofactors

Targeted LC–MS/MS analysis was performed to evaluate changes in metabolic pathways. The assessment included glycolysis, the TCA cycle, and the pentose phosphate pathway. After treating B16F10 and A375 cells with KN510713 for 24 h and comparing them to controls, ATP levels dropped by 63% in B16F10 cells, while AMP levels increased ninefold. In A375 cells, ATP levels decreased by 22%, and AMP levels rose 1.5 times (Figure 2A,B). The acylcarnitine profile was also analyzed in B16F10 and A375 cells treated with KN510713 for 24 h compared to control groups. Changes in short-chain acylcarnitine (SCAC, C2–C4), medium-chain acylcarnitine (MCAC, C6–C12), and long-chain acylcarnitine (LCAC, C14 and above) were examined. In B16F10 cells, both short- and long-chain acylcarnitine levels are increased, while in A375 cells, there was a general rise across all chain lengths, from short to long. All metabolites and acylcarnitine levels were normalized to total protein content measured using the BCA assay (Figure 2C,D).

### 2.3. FAO Inhibition Induced Synergistic Cell Death with Dacarbazine in Melanoma with BRAF Wild Type

Braf wild-type melanoma cells (B16F10) were treated with Dacarbazine (200 µM) for 0–72 h to examine molecular changes. Immunoblotting results showed that p-mTOR expression increased approximately 5.6-fold at 72 h compared to 0 h, and LC3-II expression also increased, confirming activation of autophagy during Dacarbazine treatment (Figure 3A). Before evaluating the combined effect, we verified whether CAC and ACAA1, targets of KN510 and KN713, respectively, actually contribute to FAO and ATP production [27,28]. After siRNA knockdown of *CAC* or *ACAA1* in UACC-257 melanoma cells, OCR analysis revealed a significant reduction in basal respiration and ATP production compared to the control group (Appendix A). Then, co-treatment of B16F10 cells with Dacarbazine and the FAO inhibitors KN510713 (a combination of KN510 200 µM and KN713 2 mM) resulted in a time-dependent decrease in p-mTOR and Cyclin D1 expression. Specifically, at 72 h, p-mTOR decreased by approximately 60%, and Cyclin D1 decreased by 90% compared to the control. Simultaneously, γ-H2AX expression increased significantly, indicating substantial DNA damage during the combined treatment (Figure 3B). Cell death was assessed via Annexin V/PI staining. Cell death was quantified by summing Annexin V^+^/PI^−^ (early apoptosis) and Annexin V^+^/PI^+^ (late apoptosis). FACS analysis showed that combined Dacarbazine and KN510713 treatment increased cell death by 12.6-fold compared to control and by 3-fold compared to Dacarbazine alone (Figure 3C). Furthermore, γ-H2AX immunofluorescence staining revealed a marked increase in nuclear DNA damage foci in the combination-treated group (Figure 3D). These findings suggest that FAO inhibitors enhance the anticancer effects of Dacarbazine in WT melanoma B16F10 cells, closely linked to mTOR signaling inhibition, increased DNA damage, and heightened cell death.

### 2.4. FAO Inhibition Induced Synergistic Cell Death with Dabrafenib in Melanoma with BRAF^V600E^

A375 melanoma cells with the *BRAF^V600E^* were treated with Dabrafenib (50 nM) for 0–72 h to examine molecular changes. Immunoblotting results showed that p-mTOR expression increased approximately 3.7-fold at 72 h compared to 0 h, and LC3-II expression also rose, confirming that autophagy was activated during Dabrafenib treatment (Figure 4A). We have selected inhibitors of CAC and ACAA1 as KN510 (Omeprazole) and KN713 (Trimetazidine), and the FAO inhibitory effect has been measured, as shown in Appendix A. The details of why CAC and ACAA1 are selected in FAO pathways have been reviewed recently [29]. KN510713 (a combination of 100 µM KN510 and 1 mM KN713) was administered in vitro to evaluate its combined effect with the FAO inhibitor. In A375 cells, p-mTOR and Cyclin D1 expression decreased over time. Specifically, at 72 h, p-mTOR expression decreased by approximately 70%, and Cyclin D1 was completely absent compared to the control. At the same time, γ-H2AX expression increased significantly, indicating that DNA damage was markedly induced during the combination treatment (Figure 4B). Cell death was measured using Annexin V/PI staining, analyzing the populations of Annexin V^+^/PI^−^ (early apoptosis) and Annexin V^+^/PI^+^ (late apoptosis) together. FACS results showed that the combined Dabrafenib and KN510713 treatment led to about a 9.7-fold increase in cell death compared to the control and approximately a 1.9-fold increase compared to Dabrafenib alone (Figure 4C). Additionally, γ-H2AX immunofluorescence staining revealed a significant rise in nuclear DNA damage foci in the combination group (Figure 4D). These findings suggest that FAO inhibitors enhance the anticancer effects of Dabrafenib in *BRAF^V600E^* mutant melanoma A375 cells, closely linked to mTOR signaling suppression, increased DNA damage, and heightened cell death.

**Figure 3 ijms-26-09873-f003:**
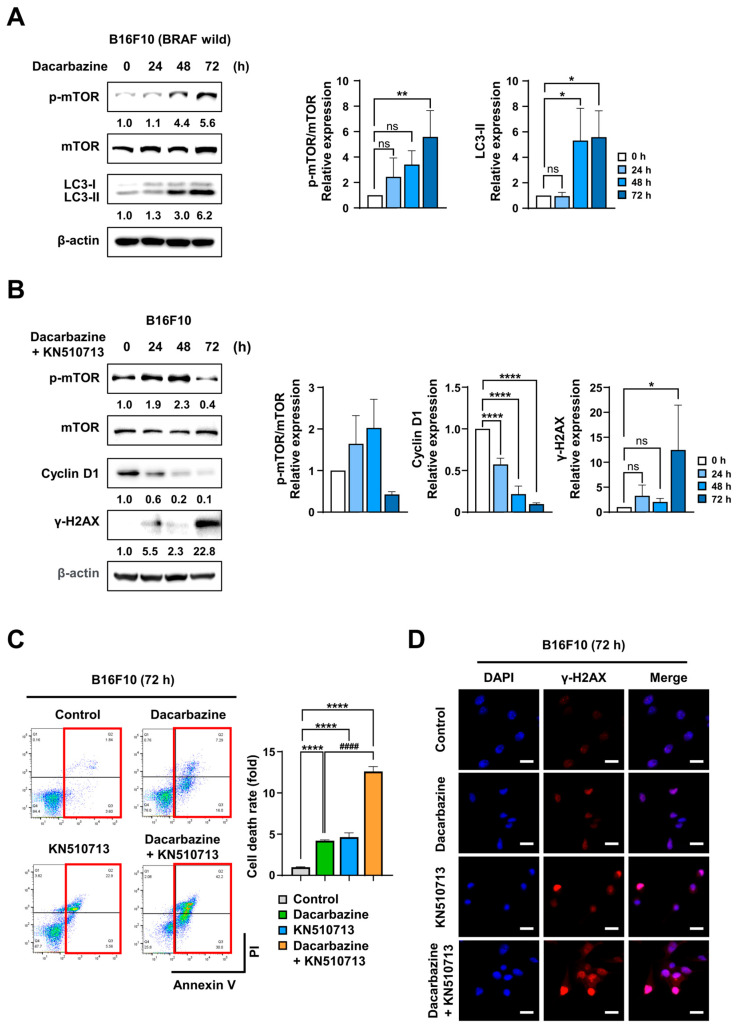
Co-treatment with Dacarbazine and KN510713 induces a cell death storm in B16F10 cells. (**A**) Immunoblotting of p-mTOR and LC3-II in B16F10 cells treated with Dacarbazine (200 µM) for 0–72 h. (**B**) Immunoblotting of p-mTOR, Cyclin D1, and γ-H2AX in B16F10 cells treated with either Dacarbazine (200 µM), KN510713, or both for 0–72 h. (**C**) FACS analysis of apoptosis using Annexin V/PI staining after 72 h. Cell death was calculated as the sum of Q2 (Annexin V^+^/PI^+^, late apoptosis) and Q3 (Annexin V^+^/PI^−^, early apoptosis) (indicated by red boxes). FACS plots are color-coded: control (gray), Dacarbazine (light green), KN510713 (light blue), and combination (orange). * *p* < 0.05, ** *p* < 0.01, and **** *p* < 0.0001 vs. Control; ^####^ *p* < 0.0001 vs. Dacarbazine; ns, not significant. (**D**) Immunofluorescence staining of γ-H2AX in B16F10 cells treated with Dacarbazine alone, KN510713 alone, or both. Nuclear DNA damage foci significantly increased with combination treatment compared to single agents (scale bars: white, 20 µm; blue).

**Figure 4 ijms-26-09873-f004:**
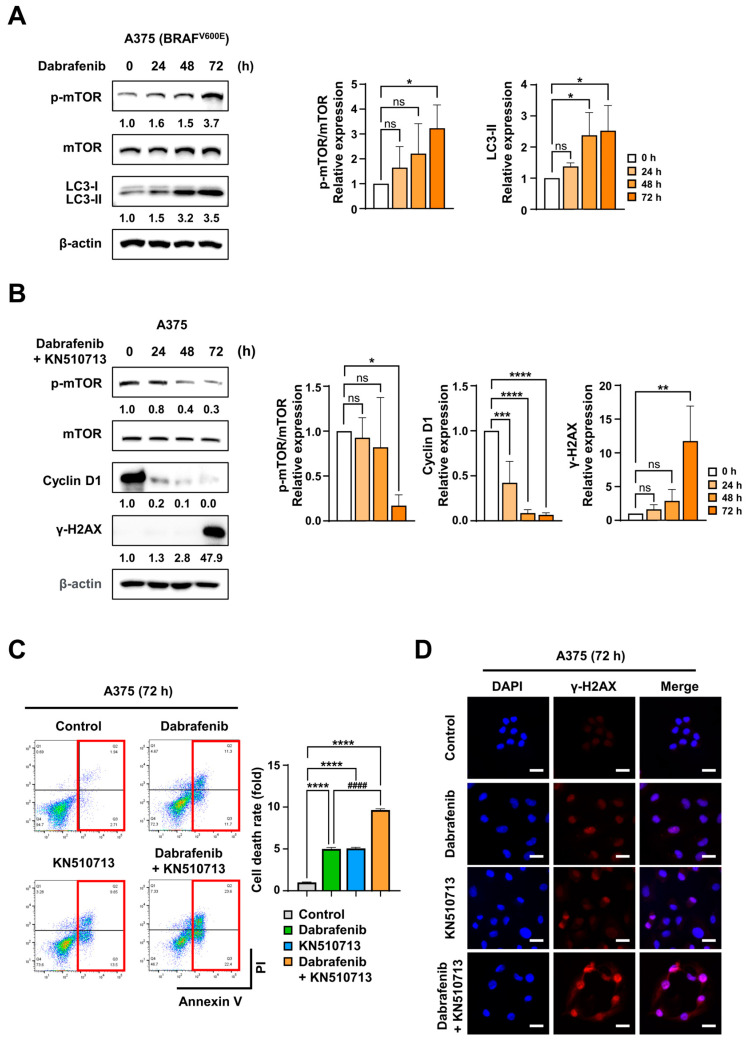
Co-treatment with Dabrafenib and KN510713 induces a cell death storm in A375 cells. (**A**) Immunoblotting of p-mTOR and LC3-II in A375 cells treated with Dabrafenib(50 nM) for 0–72 h. (**B**) Immunoblotting of p-mTOR, Cyclin D1, and γ-H2AX in A375 cells treated with Dabrafenib (50 nM) alone, KN510713 alone, or the combination of Dabrafenib and KN510713 for 0–72 h. (**C**) FACS analysis of apoptosis by means of Annexin V/PI staining after 72 h. Cell death was quantified as the sum of Q2 (Annexin V^+^/PI^+^, late apoptosis) and Q3 (Annexin V^+^/PI^−^, early apoptosis) (indicated by red boxes). FACS plots are color-coded as follows: Control (grey), Dacarbazine (light green), KN510713 (light blue), Combination (orange). * *p* < 0.05, ** *p* < 0.01, *** *p* < 0.001, and **** *p* < 0.0001 vs. Control; ^####^ *p* < 0.0001 vs. Dabrafenib; ns, not significant. (**D**) Immunofluorescence staining of γ-H2AX in A375 cells treated with Dabrafenib alone, KN510713 alone, or the combination of Dabrafenib and KN510713. Nuclear DNA-damage foci were strongly increased under combination treatment compared with single-agent therapies (scale bars: white, 20 µm; blue).

### 2.5. FAO Inhibition Induced Synergistic Cell Death with Dacarbazine in Xenograft Using Braf Wild-Type Melanoma

Colony formation assays were performed to evaluate the long-term impact of KN510 and KN713 on B16F10 cells. After seeding 1000 cells per well in a 12-well plate, KN510 (0, 12.5, 50, 200 µM) or KN713 (0, 125, 500, 2000 µM) was applied once the cells attached, and the treatment lasted for 10 days. After this period, colonies were fixed, stained, and the dye was eluted to measure absorbance. Growth inhibition was derived from these absorbance measurements to determine the IC_50_ values. In B16F10 cells, KN510 showed an IC_50_ of 21.3 µM, while KN713 had an IC_50_ of 78.4 µM (Figure 5A). Additionally, when IC_50_ was defined as the concentration that inhibits intracellular acetyl-CoA production by 50%, KN510’s IC_50_ was 10.7 µM, and KN713’s was 60.4 µM. When based on reducing ATP production by 50%, KN510’s IC_50_ was 4.6 µM, and KN713’s was 94.5 µM (Appendix A).

To assess the anticancer effects of combined Dacarbazine and FAO inhibitors in vivo, an allograft model was established by implanting B16F10 cells into C57BL/6 mice. KN510 and KN713 were given orally at 25 mg/kg daily in a 1:1 ratio, while Dacarbazine was administered intraperitoneally at 26 mg/kg daily for three weeks. Following transplantation of 1 × 10^6^ B16F10 cells per mouse (*n* = 7), tumors in mice treated with dacarbazine alone began to regrow starting at 2 weeks, indicating the development of acquired drug resistance. The combination treatment resulted in approximately 93% tumor growth inhibition compared to the control group and about 84% reduction relative to the Dacarbazine-only group at 3 weeks (Figure 5B). Body weight changes were monitored in B16F10 allograft mice during treatment. The group receiving the combined dacarbazine and KN510713 showed no change in body weight, whereas the control group exhibited an approximately 25% increase. This is likely due to the control tumor size increasing by roughly 36.6-fold, which is 13.6 times larger than the tumor size in the group treated with dacarbazine and KN510713. When comparing the body weights of the dacarbazine monotherapy group and the group treated with dacarbazine plus KN510713, the difference was within 10%. Therefore, no severe toxicity was observed with the combination therapy (Figure 5C).

### 2.6. FAO Inhibition Induced Synergistic Cell Death with Dabrafenib in Xenograft Models Using Melanoma with BRAF^V600E^

Colony formation assays were performed to evaluate the long-term effects of KN510 and KN713 on A375 cells. After seeding 1000 cells per well in a 12-well plate, KN510 (0, 12.5, 50, 200 µM) or KN713 (0, 125, 500, 2000 µM) was added for 10 days following attachment. After 10 days, colonies were fixed, stained, and the dye was eluted for absorbance measurement. Growth inhibition was calculated from these absorbance values to determine the IC_50_. In A375 cells, the IC_50_ was 96.6 µM for KN510 and 1241.4 µM for KN713 (Figure 6A). Additionally, when IC_50_ was defined as the concentration that inhibits intracellular acetyl-CoA production by 50%, the IC_50_ of KN510 in A375 cells was 10.5 µM, and that of KN713 was 60.6 µM. Furthermore, when IC_50_ was based on the concentration that reduces ATP production by 50%, the IC_50_ for KN510 was 5.3 µM, and for KN713, it was 61.2 µM (Appendix A).

To assess the anticancer effects of combined Dabrafenib and FAO inhibitors in vivo, a xenograft model was created by implanting A375 cells into BALB/c nude mice. KN510 and KN713 were administered orally once daily at 25 mg/kg each (1:1 ratio), along with oral Dabrafenib at 30 mg/kg. After transplanting 5 × 10^6^ A375 cells per mouse (*n* = 7), tumors in mice treated with Dabrafenib alone began to regrow starting at 4 weeks, indicating the development of acquired drug resistance. The combination treatment achieved nearly 96% reduction in tumor growth compared to the control group, with an approximately 83% reduction relative to the Dabrafenib-only group at 6 weeks (Figure 6B). Body weight changes were monitored in A375 xenograft mice during treatment. The weight difference among all groups was observed to be within 10% at each week. Therefore, no severe toxicity was observed with the combination therapy (Figure 6C).

**Figure 5 ijms-26-09873-f005:**
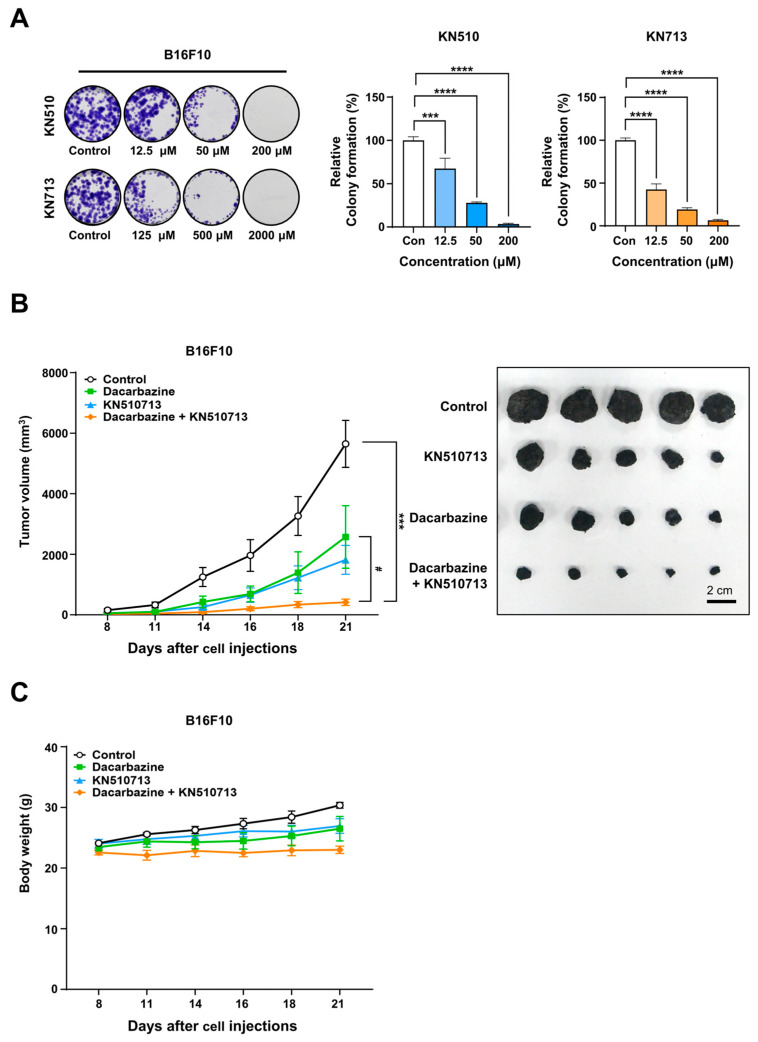
In allografts with B16F10 cells, combination treatment with Dacarbazine and KN510713 resulted in near-complete remission. (**A**) Colony formation assay of B16F10 cells treated with KN510713 (0, 12.5, 50, 200 µM) for 10 days. Colonies were fixed, stained, and the bound dye was eluted for absorbance measurement. IC_50_ values were determined based on the growth inhibition calculated from absorbance measurements. (**B**) In vivo antitumor efficacy of Dacarbazine combined with KN510713 in a B16F10 allograft model. C57BL/6 mice were implanted with B16F10 cells and treated with Dacarbazine, KN510713, or a combination of both (Dacarbazine + KN510713). Tumor growth inhibition was compared among the groups. Line colors: Control (black), Dacarbazine (light green), KN510713 (light blue), and the combination (orange). *** *p* < 0.001, **** *p* < 0.0001 vs. Control; ^#^ *p* < 0.05 vs. Dacarbazine. (**C**) Body weight changes in B16F10 allograft mice during treatment. C57BL/6 mice were implanted with B16F10 cells and treated with Dacarbazine, KN510713, or a combination of both (Dacarbazine + KN510713). Body weight was measured on days 8, 11, 14, 16, 18, and 21 after cell injection (line colors are identical to those in panel B).

**Figure 6 ijms-26-09873-f006:**
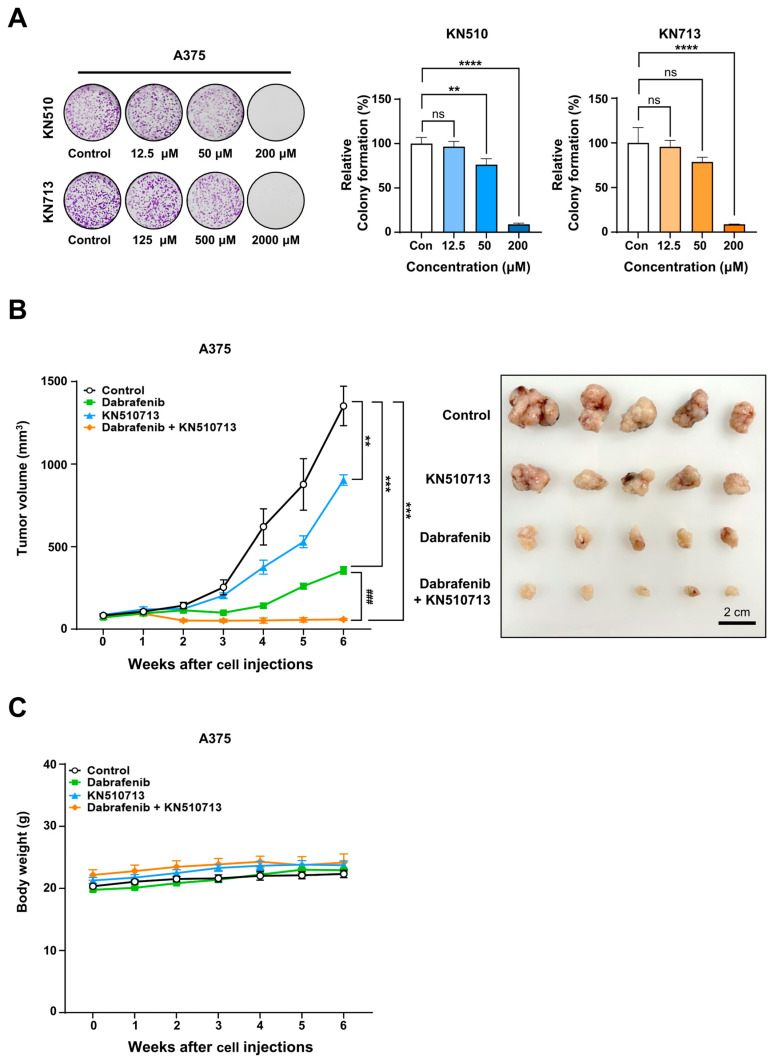
Combination treatment with Dabrafenib and KN510713 in A375 xenografts resulted in near-complete remission. (**A**) Colony formation assay of A375 cells treated with KN510713 (0, 12.5, 50, 200 µM) for 10 days. Colonies were fixed, stained, and the bound dye was eluted for absorbance measurement. IC_50_ values were calculated based on growth inhibition derived from absorbance. (**B**) In vivo antitumor efficacy of Dabrafenib combined with KN510713 in an A375 xenograft model. BALB/c nude mice were implanted with A375 cells and treated with Dabrafenib, KN510713, or the combination (Dabrafenib + KN510713). Tumor growth inhibition was compared among the groups. Line colors: Control (black), Dabrafenib (light green), KN510713 (light blue), Combination (orange). ** *p* < 0.01, *** *p* < 0.001, and **** *p* < 0.0001 vs. Control; ^###^ *p* < 0.001 vs. Dabrafenib; ns, not significant. (**C**) Body weight changes in A375 xenograft mice during treatment. BALB/c nude mice were implanted with A375 cells and treated with Dabrafenib, KN510713, or the combination (Dabrafenib + KN510713). Body weight was measured weekly from 0 to 6 weeks after cell injection (line colors are identical to those in panel B).

## 3. Discussion

In recent years, new treatment options for melanoma have expanded to include immune checkpoint inhibitors. One such option is relatlimab, a human IgG4 monoclonal antibody that blocks lymphocyte activation gene 3 (*LAG3*). It was evaluated in a Phase 2/3 clinical trial in patients with untreated metastatic melanoma in combination with nivolumab [30]. The efficacy of this new combination did not show a significant improvement in overall survival compared to established anti-PD-1/anti-CTLA-4 therapy. Recently, reports have indicated that patients with high levels of fatty acid transporters and FAO genes are associated with lower survival rates in melanoma [22,23]. Increased levels of FAO genes are involved in both peroxisomal and mitochondrial FAO [22,23]. Expression analysis in circulating tumor cells (CTCs) revealed high levels of FAO-related genes [22]. CTCs upregulate the intracellular fatty acid (FA) transport machinery, including regulators of mitochondrial membranes (CPT1A, CPT1C, CPT2, and SLC25A20) and peroxisomal membrane transport (ABCD1 and ABCD3), as well as carnitine O-octanoyltransferase (CROT) and carnitine acetyltransferase (CRAT). These are essential for managing fatty acid shuttling between peroxisomes and mitochondria. This suggests that FAO in CTCs is vital for adaptation within the bloodstream environment, potentially offering therapeutic opportunities to inhibit metastasis by blocking FAO. Knockdown of FAO-related genes, such as *CROT* or *CRAT*, in a melanoma cell line led to a significant delay in metastasis in the xenograft model [22]. Furthermore, melanoma cells resistant to BRAF/MEK inhibition showed high expression of FAO genes like acyl-CoA oxidase 1 (ACOX1), short-chain acyl-CoA dehydrogenase (ACADS), CD36, carnitine palmitoyltransferase 1A (CPT1A), acyl-CoA thioesterase 8 (ACOT8), CRAT, and enoyl-CoA hydratase and 3-hydroxyacyl-CoA dehydrogenase (EHHADH) [23]. Among these, ACOX1, CRAT, and EHHADH play key roles in peroxisome FAO [29]. The combination of BRAF/MEK inhibitors with the peroxisome proliferator-activated receptor (PPAR) agonist, thioridazine, produces a synergistic anti-tumor response in the xenograft model [23]. Unlike normal cells, melanoma maximizes fatty acid oxidation efficiency by utilizing both mitochondrial and peroxisomal pathways for FAO. This indicates that while normal cells rely on mitochondria to oxidize only long-chain fatty acids when energy demand is high, melanoma cells use a broad range of fatty acids—including short-chain, medium-chain, long-chain, and branched-chain—to support fatty acid oxidation [29]. The main limitation of the two studies mentioned above [31,32] is that they did not analyze the increase in FAO in melanoma during anticancer drug administration. They observed that FAO increased in melanoma cells that had developed resistance to anticancer drugs and demonstrated that inhibiting FAO could partially overcome this resistance. This suggests that only cells with high levels of FAO can survive during anticancer drug treatment. Even melanoma cells with low FAO levels can adapt to become cells with high FAO levels after prolonged exposure to anticancer drugs, thereby acquiring resistance. Therefore, FAO co-treatment with anticancer drugs is not limited to tumors with high FAO levels but is a necessary co-treatment for all melanoma cases. We demonstrated that, regardless of *BRAF* mutation, FAO is induced by drug treatment in this study. Additionally, despite observing that FAO acts complementarily with mitochondria and the peroxisome pathway, it is regrettable that blocking only one pathway, rather than both, to achieve anticancer effects has been reported [31,32]. Since both FAO pathways are active in melanoma, blocking both mitochondria and peroxisome FAO is necessary to optimize the FAO-inhibitory effect. We carried out experiments blocking both FAO pathways and achieved excellent synergistic anticancer effects in this study. Based on existing reports and our findings, here are the metabolic characteristics of melanoma from the perspective of cancer energy metabolism. Melanoma compensates for energy deficiency caused by glycolysis through FAO. Especially during metastasis, it requires more ATP, resulting in an increased demand for FAO [31]. To meet this increased demand for FAO, both mitochondria and peroxisomes work together to activate FAO [31,32]. When melanoma is treated with chemotherapeutic or cytotoxic agents, regardless of its mutations, it increases FAO and ATP production to survive and grow. Therefore, melanoma cells that survive anticancer drug treatment exhibit enhanced metastatic potential or drug resistance, both of which are associated with higher expression of FAO-related genes. We found that treating melanoma with both chemotherapy and FAO inhibitors simultaneously led to a 92% reduction in tumor growth compared to the control group. Consequently, to overcome drug resistance in melanoma, it is crucial to combine FAO inhibitors with either targeted therapy or chemotherapy as part of the primary treatment plan to maximize therapeutic effectiveness.

We believe that activating FAO through autophagy is linked to an increase in fatty acids caused by lipophagy [33]. Autophagy is a conserved quality control process that breaks down cell cytoplasmic contents through lysosomes. Lipophagy is a specific type of autophagy that involves the degradation of lipid droplets within cells [33]. Lipophagy releases free fatty acids, which stimulate FAO. Free fatty acids act as ligands that bind to peroxisome proliferator-activated receptors (PPAR), upregulating FAO gene expression. These transcription factors form homo- or heterodimers and move to the nucleus, where they regulate gene expression for proteins essential to FAO [34]. The final product of FAO is acetyl-CoA, which is processed in the TCA cycle to generate reductive cofactors like NADH and FADH2, followed by an electron transfer process for ATP production [35]. This increased ATP level through FAO activation helps maintain survival against chemical stress by raising levels of cyclin D1 and PARP or by directly activating mTOR [28]. We have demonstrated that *CAC* knockdown significantly reduces ATP production in various cancer cell lines grown in nutrient-rich conditions by inhibiting FAO [28]. FAO in peroxisomes is also highly active in cancer cells [29]. Therefore, inhibiting ACAA1, either through the use of an ACAA1 inhibitor [35] or by *ACAA1* knockdown, also significantly reduces ATP production under nutrient-rich conditions [27].

Therefore, regardless of *BRAF* mutations, treatment with targeted therapies or chemotherapeutic agents for melanoma increases FAO as a defense mechanism, thereby increasing ATP production and significantly supporting melanoma survival. Consequently, combining anticancer agents with FAO inhibitors, such as KN510713, is expected to produce near-complete remission therapeutic results in melanoma xenograft models.

In summary, B16F10 melanoma (Raf wild type) exhibited recurrent growth due to acquired resistance after two weeks of Dacarbazine treatment. In contrast, combination treatment with Dacarbazine and KN510713 induced near-complete remission without recurrence. A375 melanoma (*RAF^V600E^*) exhibited resistance after 4 weeks of Dabrafenib treatment, but combination therapy with Dabrafenib and KN510713 induced near-complete remission without signs of recurrence. Based on these findings, combining FAO inhibitors with first-line therapies may be a promising approach for managing melanoma, regardless of RAF mutation status.

## 4. Materials and Methods

### 4.1. Immunohistochemical Staining

Immunohistochemistry (IHC) was performed on formalin-fixed, paraffin-embedded (FFPE) melanoma tissue microarray (TMA) sections (ME2081, Tissue Array, Derwood, MD, USA). Sections were deparaffinized in xylene, rehydrated through graded ethanol, and subjected to antigen retrieval in 10 mM sodium citrate buffer containing 0.05% Tween-20 (pH 6.0) at 95 °C for 5 min. Endogenous peroxidase activity was blocked with 0.03% hydrogen peroxide, followed by blocking nonspecific binding with 2% bovine serum albumin (BSA) in Tris-buffered saline with 0.1% Triton X-100 (TBST, pH 7.6) for 30 min. Sections were incubated with the primary antibody for 1.5 h at room temperature, developed with 3,3-diaminobenzidine (DAB) for 5–20 min, counterstained with Mayer’s hematoxylin for 30–60 s, then dehydrated, cleared in xylene, and mounted. The TMA samples were obtained commercially and anonymized; therefore, IRB approval was not required. The primary antibody used was anti-ACAA1 (PA5-29956, Thermo Fisher Scientific, Waltham, MA, USA; 1:200).

### 4.2. Image Acquisition and Analysis

The stained tissue sections were scanned in high-resolution mode using the Vectra Polaris multispectral imaging system (Akoya Biosciences, Marlborough, MA, USA). Protein expression was evaluated based on the pathological assessment of tissue morphology and staining patterns in melanoma tissues. The inForm Image Analysis Software version 2.6.0 (Akoya Biosciences, Marlborough, MA, USA) was used to analyze the IHC results quantitatively. H-scores were calculated from the percentage of positively stained cells and staining intensity. The significance of protein expression across different groups was assessed using one-way ANOVA in GraphPad Prism version 10.3.1.

### 4.3. Gene Expression Analysis (GEPIA2)

Gene expression profiling was conducted using the GEPIA2 web server (accessed on 12 August 2025) [36], which integrates RNA sequencing data from The Cancer Genome Atlas (TCGA) and the Genotype-Tissue Expression (GTEx) projects. The “Expression DIY” analysis module was used to compare the mRNA expression levels of selected fatty acid oxidation (FAO)-related genes between skin cutaneous melanoma (SKCM) and normal skin tissues. Box plots were generated using the default log_2_(TPM + 1) normalization method with the following thresholds: log_2_ fold change (log_2_FC) cutoff = 0.5 and *p*-value cutoff = 0.05. Statistical significance was determined using a Student’s t-test integrated within the GEPIA2 pipeline.

### 4.4. Cell Culture and Transfection

Melanoma cell lines, including B16F10 (CRL-6475), A375 (CRL-1619), and SK-MEL-5 (HTB-70), were obtained from the American Type Culture Collection (ATCC, Manassas, VA, USA). UACC-257 cells were provided by the National Cancer Institute under a Material Transfer Agreement (No. 2702-09). All cell lines were authenticated through short tandem repeat (STR) profiling and confirmed negative for Mycoplasma, Sendai virus, and mouse hepatitis virus via PCR analysis. Cells were cultured in Dulbecco’s Modified Eagle Medium (DMEM; Cytiva, Logan, UT, USA) supplemented with 10% fetal bovine serum (FBS; Cytiva) and maintained at 37 °C in a humidified atmosphere with 5% CO_2_. For gene silencing, cells were transfected with small interfering RNAs (siRNAs) targeting CAC and ACAA1 (40 nM each) using Lipofectamine™ 3000 reagent (Thermo Fisher Scientific, Waltham, MA, USA; Cat. No. L3000075) according to the manufacturer’s protocol. After 72 h of transfection, knockdown efficiency was verified via immunoblotting. Subsequently, oxygen consumption rate (OCR) was measured to assess changes in ATP production related to fatty acid oxidation (FAO) activity.

### 4.5. Oxygen Consumption Rate (OCR) Analysis

To assess changes in oxygen consumption and ATP production following increased fatty acid uptake and inhibition of FAO, OCR was measured according to the established protocol [37]. Seahorse XFe96/XF Pro Cell Culture Microplates (Agilent Technologies, Santa Clara, CA, USA; Cat. No. 103794-100), Seahorse XF Cell Mito Stress Test Kit (Agilent Technologies; Cat. No. 103015-100), and Seahorse XF Calibrant Solution (Agilent Technologies; Cat. No. 100840-100). To normalize cell numbers per well, the Sulforhodamine B (SRB) assay was employed [38]. The experiments were performed in triplicate, and statistical significance between the control and experimental groups was analyzed using one-way analysis of variance (ANOVA) with GraphPad Prism version 10.3.1.

### 4.6. Preparation of BSA-Conjugated Palmitic Acid

Palmitic acid (Sigma-Aldrich, St. Louis, MO, USA; Cat. No. P5585) was dissolved in absolute ethanol to prepare a 200 mM stock solution. Fatty acid-free bovine serum albumin (FFA-BSA; Sigma-Aldrich; Cat. No. A0281) was prepared as a 10% (*w*/*v*) solution in Dulbecco’s Modified Eagle Medium (DMEM). The palmitic acid stock solution was slowly added to the 10% FFA-BSA solution at a final concentration of 4 mM, followed by incubation at 37 °C with gentle shaking for 1 h to allow conjugation. After incubation, the BSA–palmitate conjugate solution was filtered through a 0.22 µm membrane filter and stored at −20 °C until use.

### 4.7. Liquid Chromatography–Tandem Mass Spectrometry (LC-MS/MS)

Cells were washed sequentially with PBS and water, then metabolites were extracted with cold methanol/H_2_O (80:20, *v*/*v*). After vortexing, ^13^C_5_-Gln was added as an internal standard, and metabolites were extracted via liquid–liquid extraction with chloroform. The aqueous phase was dried via vacuum centrifugation, reconstituted in 50% methanol, and analyzed via LC–MS/MS using an Agilent 1290 HPLC coupled to a QTRAP 5500 (AB Sciex, Toronto, ON, Canada) with a Synergi fusion RP column (50 × 2 mm). The mobile phases were 5 mM ammonium acetate in water (A) and methanol (B). A gradient was applied, and metabolites were quantified through multiple reaction monitoring in negative ion mode. Peak areas were normalized to internal standards and total protein content. Data were processed with Analyst 1.7.1.

### 4.8. Immunoblotting

Cells were lysed in radioimmunoprecipitation assay (RIPA) buffer, and protein concentrations were measured with the bicinchoninic acid (BCA) protein assay kit (Pierce, Rockford, IL, USA). Equal amounts of total protein (10–40 μg per lane) were separated via SDS-polyacrylamide gel electrophoresis (SDS-PAGE) and transferred to polyvinylidene fluoride (PVDF) membranes. The membranes were blocked for 1 h at room temperature with 5% bovine serum albumin (BSA) in Tris-buffered saline (TBS) containing 0.1% (*v*/*v*) Tween-20 (TBST). After blocking, membranes were incubated overnight at 4 °C with primary antibodies. The following antibodies were used: SLC25A20 (Abcam, Cambridge, UK; Cat. No. ab244436, 1:250), ACAA1 (Thermo Fisher Scientific, Waltham, MA, USA; Cat. No. PA5-29956, 1:1000), β-actin (Santa Cruz Biotechnology, Inc., Dallas, TX, USA; Cat. No. sc-47778, 1:1000), Cyclin D1 (Cell Signaling Technology, Inc., Danvers, MA, USA; Cat. No. 55506S, 1:1000), mTOR (Cell Signaling Technology; Cat. No. 2983S, 1:1000), p-mTOR (Ser2448, Cell Signaling Technology; Cat. No. 5536S, 1:1000), LC3-I/II (Cell Signaling Technology; Cat. No. 12741S, 1:1000), and γ-H2AX (Bethyl Laboratories, Inc., Montgomery, TX, USA; Cat. No. A300-081A, 1:1000). After three washes with TBST, membranes were incubated for 1 h at room temperature with horseradish peroxidase (HRP)-conjugated secondary antibodies. The membranes were then washed five times with TBST, and protein signals were detected using Westsave™ chemiluminescent substrate (AbFrontier, Seoul, Republic of Korea). Gel images were captured with a FUSION-Solo.4.WL imaging system (Vilber Lourmat, Marne-la-Vallée, France). Densitometric analysis of immunoblot bands was performed using ImageJ software (version 1.53t; National Institutes of Health, Bethesda, MD, USA). Briefly, the grayscale Western blot images were imported into ImageJ, and each protein band was defined as a region of interest (ROI) using the rectangular selection tool. The integrated density (area × mean gray value) of each band was measured, and background intensity was subtracted using the “Rolling Ball” algorithm with a radius of 50 pixels. The relative protein expression levels were normalized to β-actin as a loading control. Normalized values were expressed as fold change relative to the control group, and statistical analyses were performed from at least three independent experiments. All quantification data are presented as the mean ± standard deviation (SD), and statistical significance was determined as described in Section 4.15.

### 4.9. Measurement of ATP Levels

Intracellular ATP levels were determined using an ATP Colorimetric/Fluorometric Assay Kit (Abcam, Cambridge, UK; Cat. No. ab83355) following the manufacturer’s instructions. Briefly, melanoma cells were lysed in the supplied ATP assay buffer, and cell lysates were deproteinized with a 10 kDa molecular-weight cut-off spin filter to remove interfering substances. The absorbance at 570 nm (colorimetric mode) was measured using a microplate reader to determine the ATP content. ATP concentrations were derived from a standard curve and normalized to total protein content, which was assessed using the BCA Protein Assay Kit (Thermo Fisher Scientific, Waltham, MA, USA).

### 4.10. Apoptosis Analysis

Apoptotic and necrotic cell populations were quantified using an Annexin V–abFluor 488/Propidium Iodide (PI) Apoptosis Detection Kit (Abbkine, Wuhan, China; Cat. No. KTA0002) according to the manufacturer’s protocol. After treatment, both adherent and floating cells were collected, washed twice with cold phosphate-buffered saline (PBS), and resuspended in 1× binding buffer at a concentration of 1 × 10^6^ cells/mL. Cells were incubated with Annexin V–abFluor 488 and PI for 15 min at room temperature in the dark. Fluorescence was measured using a BD FACSCanto™ II flow cytometer (BD Biosciences, Franklin Lakes, NJ, USA), and data were analyzed with FlowJo software (version 10.10.0, BD Biosciences). Cells were classified into four populations based on fluorescence profiles: viable (Annexin V^−^/PI^−^), early apoptotic (Annexin V^+^/PI^−^), late apoptotic (Annexin V^+^/PI^+^), and necrotic (Annexin V^−^/PI^+^).

### 4.11. Immunofluorescence

Cells grown on glass coverslips were fixed with 4% paraformaldehyde for 15 min at room temperature and permeabilized with 0.3% Triton X-100 in phosphate-buffered saline (PBS) for 10 min. After washing, cells were blocked with 3% bovine serum albumin (BSA) in PBS containing 0.05% Tween-20 (PBST) for 1 h at room temperature. Cells were then incubated overnight at 4 °C with primary antibodies, followed by 1 h incubation at room temperature with Alexa Fluor–conjugated secondary antibodies (Invitrogen, Carlsbad, CA, USA). Nuclei were counterstained with 4′,6-diamidino-2-phenylindole (DAPI, 300 nM; Invitrogen) for 5 min. After washing, coverslips were mounted using antifade mounting medium (Invitrogen), and fluorescence images were acquired using a fluorescence microscope.

### 4.12. Colony Formation Assay

To evaluate long-term cell growth, melanoma cells were seeded in 12-well plates at a density of 1 × 10^3^ cells per well and treated with KN510 (0, 12.5, 50, and 200 µM) or KN713 (0, 125, 500, and 2000 µM) for 10 days. The culture medium containing the respective compounds was refreshed every 3 days. At the endpoint, cells were washed twice with ice-cold phosphate-buffered saline (PBS) and fixed with ice-cold methanol for 10 min. Colonies were stained with 0.1% (*w*/*v*) crystal violet (Sigma-Aldrich, St. Louis, MO, USA) prepared in PBS containing 10% ethanol for 5 min, followed by gentle washing with PBS. After air drying, plates were imaged using a flatbed scanner. For quantification, the bound dye was solubilized in elution buffer (50% ethanol, 40% distilled water, and 10% acetic acid) for 5 min, and aliquots were transferred to 96-well plates. The absorbance at 580 nm was measured using a microplate reader.

### 4.13. Acetyl-CoA Assay

Intracellular acetyl-CoA levels were measured using an Acetyl-CoA Assay Kit (Abcam, Cambridge, UK; Cat. No. ab87546) following the manufacturer’s instructions. Briefly, cells were collected and homogenized in 1 N perchloric acid (PCA) and then sonicated on ice. The lysates were centrifuged at 10,000× *g* for 10 min at 4 °C, and the supernatants were deproteinized via incubation with ice-cold PCA. The deproteinized samples were neutralized to pH 7–8 with 2 M KOH, then centrifuged again to remove precipitated salts. The clarified supernatants were used for acetyl-CoA measurement according to the protocol provided with the assay kit. The absorbance at 450 nm was measured using a microplate reader.

### 4.14. Allograft and Xenograft Models

All animal experiments were approved by the Institutional Animal Care and Use Committee (IACUC) of the National Cancer Center Research Institute (protocols: NCC-24-1029, NCC-25-1162-001) and were conducted following the ARRIVE 2.0 guidelines and institutional ethical standards. In the B16F10 allograft model, C57BL/6 mice (Orient Bio, Seongnam, Republic of Korea; 6 weeks old) were subcutaneously injected with 1 × 10^6^ B16F10 cells suspended in 200 µL phosphate-buffered saline (PBS). In the A375 xenograft model, BALB/c nude mice (Orient Bio, Seongnam, Republic of Korea; 6 weeks old, female) were subcutaneously injected into the right flank with 5 × 10^6^ A375 cells suspended in 200 µL of a 1:1 mixture of PBS and Matrigel (Corning, Cat. No. 354230; supplied by BD Biosciences, Franklin Lakes, NJ, USA). Treatment was started on the day of tumor cell injection in the B16F10 model and when tumor volume reached approximately 100 mm^3^ in the A375 model. The compound KN510713 (a combination of KN510 and KN713) was administered orally once daily at 25 mg/kg in both models. In the B16F10 allograft model, dacarbazine (26 mg/kg) was administered intraperitoneally, while in the A375 xenograft model, dabrafenib (30 mg/kg) was given orally. Tumor volume (V) was calculated with the formula V = (length × width^2^)/2, and both tumor volume and body weight were measured weekly. Humane endpoints included >20% body weight loss or tumor volume exceeding 2000–3000 mm^3^, whichever occurred first.

### 4.15. Statistical Analysis

Statistical analyses were performed using GraphPad Prism 10 (GraphPad Software, San Diego, CA, USA). Comparisons between two groups were conducted with Student’s *t*-test, while one-way or two-way analysis of variance (ANOVA) was used for multiple group comparisons. Statistical significance was set at *p* < 0.05, *p* < 0.01, *p* < 0.001, and *p* < 0.0001; ns indicates ‘not significant’. In all results, statistical significance relative to the control group is marked by asterisks (*), while significance relative to the first-line standard chemotherapy group is marked by number signs (#).

### 4.16. Ethics Statement

All animal experiments were reviewed and approved by the Institutional Animal Care and Use Committee (IACUC) of the National Cancer Center Research Institute (protocols: NCC-24-1029, NCC-25-1162-001), an Association for Assessment and Accreditation of Laboratory Animal Care International (AAALAC International)-accredited facility that adheres to the Guide for the Care and Use of Laboratory Animals.

## 5. Conclusions

We found that melanoma induces autophagy regardless of *BRAF* mutation when treated with Dabrafenib or Dacarbazine. This autophagy activates FAO to produce more ATP for survival. We demonstrated that melanoma tumor regrowth by Dabrafenib or Dacarbazine treatment is completely reversed by inhibiting FAO using KN510713 (targeting CAC and ACAA1/2). Therefore, inhibiting FAO, in conjunction with standard therapy, may help overcome resistance to chemotherapy in melanoma.

## Figures and Tables

**Figure 1 ijms-26-09873-f001:**
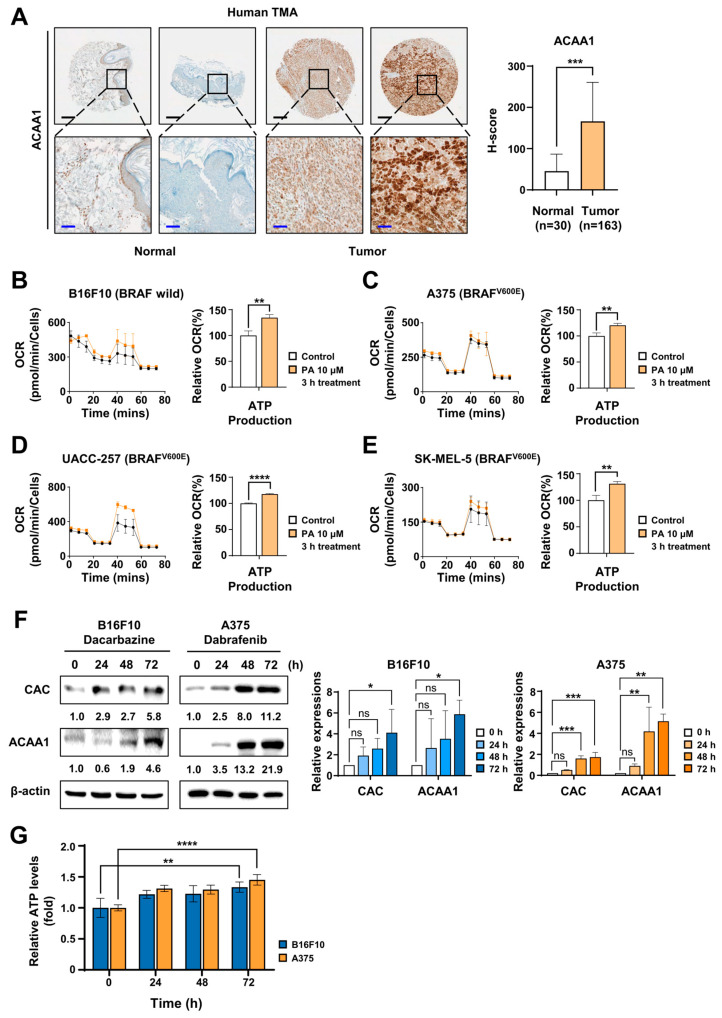
Primary anticancer drug treatment significantly stimulates fatty acid oxidation in melanoma. (**A**) Immunohistochemical staining of ACAA1 in tissue microarrays (TMAs) consisting of normal skin (*n* = 30) and melanoma tissues (*n* = 163). Representative images are shown (scale bars: black, 200 µm; blue, 50 µm). H-scores were quantified using Inform software. (**B**–**E**) Seahorse OCR analysis showing ATP production in B16F10, A375, UACC-257, and SK-MEL-5 cells treated with 10 µM BSA-conjugated palmitic acid for 3 h. ATP production was significantly higher compared to controls. (**F**) Immunoblotting of CAC and ACAA1 in B16F10 cells treated with Dacarbazine (200 µM) and A375 cells treated with Dabrafenib (50 nM) at various time points (0–72 h). Protein levels increased over time. (**G**) Intracellular ATP levels in B16F10 and A375 cells treated with Dacarbazine or Dabrafenib, respectively, over 0–72 h. Both cell lines showed a time-dependent increase in ATP production. * *p* < 0.05, ** *p* < 0.01, *** *p* < 0.001, and **** *p* < 0.0001 vs. Control; ns, not significant.

**Figure 2 ijms-26-09873-f002:**
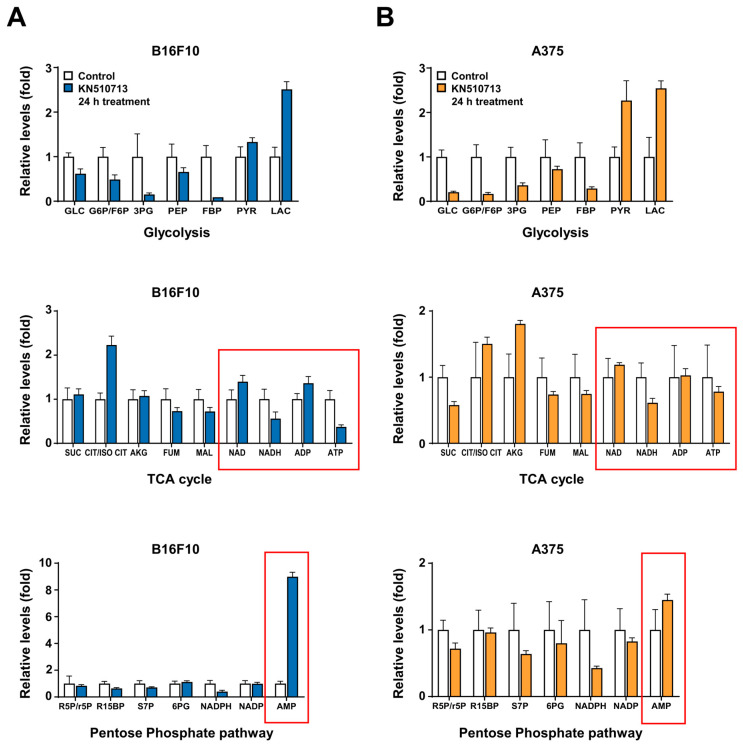
Inhibiting fatty acid oxidation increases acyl-carnitine levels and decreases the reduction of energy cofactors. (**A**,**B**) Targeted LC–MS/MS analysis of B16F10 and A375 cells treated with KN510713 for 24 h. Metabolite changes in glycolysis, the TCA cycle, and the pentose phosphate pathway were assessed, showing decreased ATP and increased AMP levels. The red box indicates the cofactors required for energy metabolism. (**C**,**D**) Acylcarnitine profiling under the same conditions. The relative abundance of short-chain (SCAC, C2–C4), medium-chain (MCAC, C6–C12), and long-chain (LCAC, C14 and above) acylcarnitines was compared between control and KN510713-treated cells. Increased levels were observed in short- and long-chain species in B16F10, while broad elevations from short- to long-chain acylcarnitines were found in A375. All metabolite levels were normalized to total protein content measured using the BCA assay.

## Data Availability

No datasets were generated or analyzed during the current study.

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
