# Peer review of "Inhibiting Fatty Acid Oxidation Suppresses Acquired Resistance to Standard Chemotherapy in Melanoma"

_ijms, 2025, doi:10.3390/ijms26209873_

Round 1

Reviewer 1 Report

Comments and Suggestions for Authors

The reviewed article is focused on the study of a promising approach to the treatment of metastatic melanoma which combines chemotherapy with Dabrafenib or Dacarbazine and fatty acid oxidation inhibitors. The authors' idea is that standard chemotherapy or immunotherapy increase autophagy in tumor cells which causes activation of fatty acid oxidation, and which in turn contributes to the development of anti-chemotherapy resistance. As a result, inhibition of fatty acid oxidation may reduce resistance and thus improve the efficacy of chemotherapy in patients with advanced melanoma.

This is described in detail in the Introduction to the article supported by modern scientific literature. Though the reference list is rather short (only 28 references) more than 50% of the publications are dated to the last 5 years. Citing of earlier works is quite justified due to their content.

The “Results” section is well structured which allows the reader to navigate easily through the research performed by the authors. Every subsection is well illustrated by relevant Figures which helps still deeper understand the logics of the study and the obtained results.

It should be noted that the authors used various and multiple modern methods during their investigation in order to reliably prove the mechanism of anti-tumor action of combined therapy with standard cytostatic agents and inhibitors of fatty acid oxidation.

The description of the used methods is presented at the end of the manuscript. It is rather short but very precise and accurate enabling the interested researchers to reproduce the experiments and obtain similar results.

Relevant statistical methods were used to prove reliability of the differences between experimental and control groups.

The “Discussion” section is written in detail using modern scientific literature to compare it with the obtained data. There are 4 references to the works of the authors of the manuscript but they cannot be considered as excessive self-citation since the presented data obtained earlier support the results presented in the reviewed article.

The conclusion presented in the manuscript comes out logically from the results of the study summarizing them.

The article is written in good scientific English but contains some inaccuracies that should be corrected.

Comments:

  1. The abstract (line 34) and the Results section (line 385) state: A549 melanoma. This is an error, as the A549 cell line is cultured non-small cell lung cancer cells, while the authors used A375 melanoma cells in their study.
  2. In the Results section (line 289), the word “reduction” or similar is missing when describing the effect compared to the control group (96%...).

The article may be recommended for publication after minor editorial revisions without re-review.

Reviewer 2 Report

Comments and Suggestions for Authors

General Assessment

This manuscript by Choi et al. investigates the role of fatty acid oxidation (FAO) in promoting drug resistance in melanoma and explores the effects of dual FAO inhibition (via KN510713) in combination with dacarbazine or dabrafenib. The study presents intriguing preclinical evidence that targeting FAO can enhance therapeutic responses and overcome resistance. The findings are potentially impactful; however, several issues need to be addressed to strengthen the mechanistic depth, translational relevance, and overall rigor before this work can be considered for publication in a high-impact venue.

Minor Comments

  • Several figures (e.g., immunoblots) would benefit from quantification (densitometry) to support claims.
  • Tumor growth curves should include body weight and toxicity data to assess tolerability.
  • The quality of the figures should be revised; when zooming in; the axis labels in several graphs are not legible.
  • In Figure 3B, it is unclear whether LC3-II decreases along with p-mTOR or remains constant under combination treatment. The Western blot should be shown explicitly.
  • In the supplementary materials, Figure S5 is mistakenly labeled under the title of Supplementary Figure 6 — this should be corrected.
